# Fyn Tyrosine Kinase Elicits Amyloid Precursor Protein Tyr682 Phosphorylation in Neurons from Alzheimer’s Disease Patients

**DOI:** 10.3390/cells9081807

**Published:** 2020-07-30

**Authors:** Filomena Iannuzzi, Rossana Sirabella, Nadia Canu, Thorsten J. Maier, Lucio Annunziato, Carmela Matrone

**Affiliations:** 1Department of Biomedicine, Aarhus University, Aarhus C, 8000 Aarhus, Denmark; filomena.iannuzzi@biomed.au.dk; 2Division of Pharmacology, Department of Neuroscience, School of Medicine, University of Naples Federico II, 80131 Naples, Italy; sirabell@unina.it; 3Department of System Medicine, University of Rome “Tor Vergata”, 00133 Rome, Italy; nadia.canu@uniroma2.it; 4Institute of Biochemistry and Cell Biology, CNR, 00015 Monterotondo, Rome, Italy; 5Paul-Ehrlich-Institut, (Federal Institute for Vaccines and Biomedicines), 63225 Langen, Germany; thorstenjuergen.maier@pei.de; 6SDN Research Institute Diagnostics and Nuclear (IRCCS SDN), Gianturco, 80131 Naples, Italy

**Keywords:** amyloid precursor protein, amyloid beta, Fyn tyrosine kinase, Tyr682 residue, YENPTY domain

## Abstract

Alzheimer’s disease (AD) is an incurable neurodegenerative disorder with a few early detection strategies. We previously proposed the amyloid precursor protein (APP) tyrosine 682 (Tyr682) residue as a valuable target for the development of new innovative pharmacologic or diagnostic interventions in AD. Indeed, when APP is phosphorylated at Tyr682, it is forced into acidic neuronal compartments where it is processed to generate neurotoxic amyloid β peptides. Of interest, Fyn tyrosine kinase (TK) interaction with APP Tyr682 residue increases in AD neurons. Here we proved that when Fyn TK was overexpressed it elicited APP Tyr682 phosphorylation in neurons from healthy donors and promoted the amyloidogenic APP processing with Aβ peptides accumulation and neuronal death. Phosphorylation of APP at Tyr (pAPP-Tyr) increased in neurons of AD patients and AD neurons that exhibited high pAPP-Tyr also had higher Fyn TK activity. Fyn TK inhibition abolished the pAPP-Tyr and reduced Aβ42 secretion in AD neurons. In addition, the multidomain adaptor protein Fe65 controlled the Fyn-mediated pAPP-Tyr, warranting the possibility of targeting the Fe65-APP-Fyn pathway to develop innovative strategies in AD. Altogether, these results strongly emphasize the relevance of focusing on pAPP Tyr682 either for diagnostic purposes, as an early biomarker of the disease, or for pharmacological targeting, using Fyn TKI.

## 1. Introduction

Alzheimer’s disease (AD) is a devastating public health problem. Although a large body of evidence indicates the detrimental role of amyloid-β (Aβ) peptide in the form of aggregates or soluble oligomers in the pathology of AD, the precise mechanisms that are responsible for Aβ accumulation remain unknown [1,2,3].

To generate Aβ peptides, the amyloid β precursor protein (APP) must first be cleaved by the aspartyl protease β-site APP-cleaving enzyme-1 (BACE-1). As BACE-1 is optimally active in acidic environments, such as in the late endosome and lysosome [4,5], it is reasonable that increased Aβ peptide production might be due to a preferential APP trafficking towards such organelles. We and others previously reported that the tyrosine 682 (Tyr682) residue in the _682_YENPTY_687_ motif of APP is crucial for APP trafficking in neurons [6,7,8,9,10,11]. Tyr682 phosphorylation regulates APP binding to specific adaptors and thus controls APP endocytosis and distribution in neurons [8,12,13,14,15]. When Tyr682 is replaced by glycine (YG), APP endocytosis is affected, and APP is processed by α-secretase on the plasma membrane to generate soluble APPα (sAPPα) fragments [16,17]. Of note, YG mice also exhibit alterations of APP subcellular trafficking [17] and deficits in autophagy, with enlarged and dysmorphic APP-enriched lysosomes [18]. Additionally, YG mice exhibit premature ageing, characterized by extensive cognitive and learning deficits that are reminiscent of those found in animal models of AD [17,19].

Our previous studies examined whether the extent of APP phosphorylation at Tyr682 is altered in AD neurons. We found that APP Tyr682 phosphorylation increased in neurons from three AD patients who carried mutations of the presenilin 1 (*PSEN1*) gene [20]. This increase was associated with impairments in APP endocytosis and forced APP accumulation in the acidic neuronal compartments, where it is processed to generate Aβ peptides [5,20]. We then searched for the tyrosine kinase (TK) that is involved in APP phosphorylation at Tyr682 in AD neurons and using mass spectrometry analysis we found that Fyn TK interacted only with this residue, and the interaction was elevated in brain tissue from AD modeling Gottingen minipigs [21] and patients both carrying *PSEN1* gene mutations [20]. Notably, Fyn TK inhibitors (TKIs) reversed the alterations in APP endocytosis and trafficking and prevented Aβ production [20]. In this regard, it should be mentioned that Fyn belongs to the Src kinase family and is enriched at synaptic structures where it regulates synaptic transmission and plasticity [22]. Importantly, Fyn expression levels increase in AD brains, where it has been proposed to regulate Aβ-mediated synaptic toxicity [23,24,25,26]. Consistently, the genetic ablation of Fyn decreased Aβ-induced synaptoxicity [27,28,29]. In addition, Fyn phosphorylates the Tyr18 residue of Tau protein in neuronal cells [30] and activates glycogen synthase kinase 3β to promote the neurotoxic Tau pathway, including Tau hyperphosphorylation at serine/threonine residues [31,32].

In this study, we investigate the possibility that Fyn overactivation promotes APP Tyr682 phosphorylation and triggers amyloidogenic APP processing in neurons from AD patients thus pointing on APP Tyr682 phosphorylation for the development of innovative strategies in AD diagnosis and treatment.

## 2. Materials and Methods

### 2.1. Human Neural Progenitors

Human induced pluripotent stem cells (iPSCs) and human neural stem cells (hNSCs) were purchased from Coriell Institute (New Orleans, LA, USA) and Axol Bioscience (Cambridge, UK), respectively. Donor clinical features are provided in Table 1, based on information from the Axol Bioscience website (https://www.axolbio.com) and CIRM Repository website (https://fujifilmcdi.com/the-cirm-ipsc-bank/).

Coriell iPSCs were induced to neurons using Gibco PSC Neural Induction Medium according to the supplier’s website and kept in culture in Neurobasal/B27 until the neuronal phenotype developed (https://www.thermofisher.com/it/en/home/references/protocols/neurobiology/neurobiology-protocols/induction-of-neural-stem-cells-from-human-pluripotent-stem-cells-using-gibco-pscneural-induction-medium.html).

For the transfection experiments, we used neural stem cells 24 h after plating (2,500,000 cell/well, well plate diameter 35 mm) in Neurobasal/B27. The experiments reported in the study were all performed in neural stem cells from the healthy volunteer #ax0019. However, experiments were also replicated in #ax0015 and #ax0018, without detecting any relevant experimental differences between the three different neural cell lines.

For immunofluorescence (IF) analysis and Aβ42 ELISA, we used neurons that were differentiated for at least 5 weeks in culture in Neurobasal/B27 medium (250,000 cells/well, well plate diameter 10 mm) following the procedure described by Zollo et al. [33].

For transfection, 500 ng/mL of each type of DNA (1 μg/mL per well) was incubated with Lipofectamine Stem Transfection Reagent for 20 min in Optimem medium. The mixture was then transferred to cells that were plated in Neurobasal/B27 medium. After 24 h, the medium was refreshed, and cells were cultured for additional 24 h.

For siRNA transfection, 5 pmol of Fyn siRNA was incubated with Lipofectamine RNAiMAX Transfection Reagent for 20 min in Optimem medium. The mixture was then transferred to cells that were plated in Neurobasal/B27 medium. After 24 h, the medium was refreshed, and cells were cultured for an additional 24 h. We used 5 pmol Silencer GAPDH siRNA as the positive control and Silencer Negative Control #1 siRNA as the negative control following the supplier’s suggestions (data not shown).

### 2.2. Immunoprecipitation (IP) Assay and Western Blot (WB) Analysis

Transfected NSCs or cultured fibroblasts were lysed in RIPA buffer in the presence of protease and phosphatase inhibitors. Protein concentrations were estimated using the Pierce BCA Protein Assay Kit. Equal amounts (20 μg) of proteins were separated on 4–20% Bis-Tris sodium dodecyl sulfate-polyacrylamide gel (SDS-PAGE) to analyze the C-terminal fragment of APP or full-length APP, respectively. For protein blotting, polyvinylidene difluoride membranes were used after activation in methanol solution. Membranes were incubated overnight with primary antibody diluted in 5% bovine serum albumin. Protein band visualization was performed using a Chemidoc MP imaging system with Image Lab software (Biorad, Copenhagen, DK).

For pTyr IP, total lysates were incubated overnight with mouse pTyr antibody (magnetic bead conjugate). The IP samples were analyzed by WB using rabbit anti-APP, rabbit anti-pan Fyn and rabbit anti-Src pTyr416 (also recognizing Fyn Tyr420).

### 2.3. Inhibitors

To investigate the effects of Fyn TKs on the levels of pAPP-Tyr682, we treated differentiated neurons from AD patients and healthy controls with five different TK inhibitors (TKIs) (Table 2). After incubation for 6 h, the neurons were washed and left in culture for an additional 12 h in Neurobasal/B27 media. The next day, the media were collected, and neurons were counted after DAPI staining to evaluate neuronal toxicity.

### 2.4. ELISA

A total of 200,000 NSCs were cultured on 24-well plates in 0.3 mL Neurobasal/B27 medium. After 5 weeks in culture, the neurons were washed in 1X phosphate-buffered saline and exposed to fresh media for 24 h. Media were collected after 48 h of incubation, centrifuged at 1000 rotations per minute for 10 min to eliminate cellular debris, and analyzed by ELISA using a Human Aβ42 ELISA kit. Aβ42 values for each sample were compared with standard curves, which were generated from samples of known concentrations of Aβ42 (0.040–2.0 ng/mL) according to the manufacturer’s instructions. The amount of Aβ42 released in the media was normalized to the total number of cells that were present on the corresponding slide and were expressed as nanograms of Aβ per single cell.

### 2.5. Materials

Neural cell culture media were from Gibco (Thermo Fisher Scientific, Copenhagen, DK).

For transfection experiments we used: Lipofectamine Stem Transfection Reagent (Thermo Fischer Scientific, Copenhagen, DK), RIPA buffer and phosphatase inhibitors (Sigma Aldrich, Søborg, DK), and 4–20% Bis-Tris sodium dodecyl sulfate-polyacrylamide gel (SDS-PAGE) (Biorad, Copenhagen, DK).

ELISA kits were from Invitrogen (#KHB3544, Thermo Fisher, Copenhagen, DK).

For siRNA experiments: Lipofectamine RNAiMAX Transfection Reagent, Silencer Negative Control #1 siRNA (#AM4611), Silencer GAPDH siRNA Positive Control (#AM4624), and Fyn siRNA (#AM51331) were from Ambion (Thermo Fisher Scientific, Milan, IT).

All the TKIs were provided by Selleckchem (Rungsted, DK).

Monoclonal anti-β-actin (#A1978) and monoclonal anti-β-actin-peroxidase (#A3854) were from Sigma Aldrich (Søborg, DK).

The antibodies that were used for WB were the following: anti-APP, clone Y188 (#ab32136, Abcam, UK), rabbit anti-pan Fyn and anti-Src pTyr416 (#4023 and #2101, Cell Signaling Technology, BioNordika, Herlev, DK), Phospho-Tyrosine Mouse mAb (P-Tyr-100) (Magnetic Bead Conjugate) (#8095, Cell Signaling Technology, BioNordika, Herlev, DK), mouse anti-pTyr clone PY20 (#P4110, Sigma Aldrich, Søborg, DK), and rabbit anti-Fe65 (#ab91650, Abcam, UK).

For IP analysis, we used anti-pTyr antibody agarose conjugated magnetic beads (#8095), rabbit anti-pan Fyn (#4023), and rabbit anti-Src pTyr416 (#2101), all provided by Cell Signaling Technology (BioNordika, Herlev, DK).

For the transfection experiments, the following plasmid vectors were used: pEGFP-n1APP (#69924, Addgene, Teddington, UK), pmApple-FYN-N-10 (#54903, Addgene, Teddington, UK), pEGFP-n1-AICD (#69925, Addgene, Teddington, UK), pCAX APP-695 (#30137, Addgene, Teddington, UK), pRK5 Fyn SH2-R176E (harboring a mutation on arginine 176 to glutamic acid on the SH2 domain; #16035, Addgene, Teddington, UK), and pRK5 Fyn deltaSH3 (harboring a deletion of the SH3 domain; #160334, Addgene, Teddington, UK). pAPP-Y682G (plasmids that harbored mutations on the Tyr682 residue to Gly) were provided by Prof. Canu. pRK5 c-Fyn and pRK5 dnFyn (K299M mutation) have been generated by Dr. Giancotti (#16032 and #16033, Addgene, Teddington, UK).

### 2.6. Statistical Analysis

The data are expressed as mean ± SEM. All of the experiments were performed at least three times (*n* = 3). The data were analyzed using GraphPad Prism 8.0c software (San Diego, CA, USA). The specific statistical tests are reported in the figure legends.

## 3. Results

### 3.1. Fyn Overexpression and Overactivation Promoted APP Amyloidogenic Processing and Neuronal Death in Human Neurons

Several studies reported a role for Fyn in AD and AD-related disorders [34,35]. However, to our knowledge, the role of Fyn in mediating APP phosphorylation at Tyr682 and triggering AD-like processes in human neurons has not yet been reported. To elucidate the interaction between APP and Fyn and determine whether Fyn plays a role in APP phosphorylation at Tyr, neural stem cells (NSCs) from healthy donors were transfected with C-terminal green fluorescent protein (GFP) tagged APP and C-terminal apple tagged Fyn. Control cells were transfected with a GFP empty vector. The expression levels of APP-GFP and Fyn-apple were first visualized and quantified by WB 48 h post-transfection (Figure 1A,B). The appearance of bands at approximately 150 kDa and 85 kDa corresponding to the molecular weight of APP and Fyn tagged fused protein, respectively, indicated that the neurons were properly transfected (Figure 1A,B). Of note, WB using the anti-GFP antibody clearly showed, in addition to the full-length GFP tagged APP protein, the appearance of the APP intracellular domain (AICD) fragment (approximately 33 kDa) in APP-GFP+Fyn co-transfected samples, indicating an increase in APP cleavage (Appendix A). As an internal control to confirm that the 33 kDa band corresponded to the AICD fragment, neurons were transfected with AICD-GFP-tagged DNA. As expected, the transfected AICD-GFP fragment migrated at the same band size that was observed for APP+Fyn neurons (Appendix A). Interestingly, a large extent of APP-GFP+Fyn-apple transfected neurons were not viable after 48 h of transfection (Figure 1D). The same toxicity was not detected in neurons transfected with APP-GFP, Fyn-apple or the GFP empty control vector, suggesting that Fyn, when overexpressed with APP, triggers molecular events that ultimately result in neuronal toxicity. Notably, a large number of cells appeared to be transfected with APP-GFP (green), Fyn-apple (red), and APP-GFP+Fyn-apple (yellow) when compared with the total number of DAPI stained nuclei (75% ± 11.3% APP-GFP vs. 100% ± 11.8% DAPI stained nuclei; 67% ± 6.9% Fyn-apple vs. 100% ± 9.5% DAPI stained cells; and 65% ± 9.7% APP-GFP+Fyn-apple vs. 100% ± 8.9% DAPI).

To determine whether the increase in APP processing to generate the intracellular AICD fragment as well as the extensive neuronal death that was observed in APP+Fyn neurons were associated with Aβ42 release, we performed an ELISA on media from control (Ctrl), APP, Fyn, and APP+Fyn neurons (Figure 1C). We observed an increase in Aβ42 levels in media from APP+Fyn neurons compared with APP neurons, suggesting that Fyn overexpression promotes amyloidogenic APP processing in transfected neurons. Consistently, APP overexpression alone fails in increasing Aβ42 levels and neuronal death, suggesting that other factors, such as Fyn overactivation, are necessary to initiate the amyloidogenic pathway. Relevantly, the absence of β-actin as well as of full-length APP in media make it extremely unlikely that Aβ is released by neurons after death (data not shown).

### 3.2. Fyn Overactivation Failed in Phosphorylating APP When Tyr682 Was Replaced by Gly in Human Neurons

Fyn TK has been previously reported to be responsible for the Tyr phosphorylation of proteins that are involved in AD progression and AD-like neurodegenerative events [34,36,37]. We examined whether Fyn is involved in APP phosphorylation at Tyr682 in human neurons. APP, Fyn, and APP+Fyn untagged-DNA were transfected in neuronal stem cells, and after 48 h cells were collected and total lysates immunoprecipitated with the anti-pTyr antibody coupled with magnetic beads and analyzed with anti-APP or anti-Fyn antibodies by WB (Figure 2A). We used Fyn and APP untagged instead of Fyn-apple and APP-GFP in these experiments to exclude that APP-Fyn crosstalk (see Figure 1) may be influenced by the presence of tags (Figure 2A). Basal expression levels of total APP, total Fyn, and pFyn-Tyr420 were assessed on total lysates by WB (Figure 2C,D). pAPP-Tyr levels significantly increased in APP+Fyn neurons compared with APP neurons (Figure 2B). We used dominant-negative Fyn (dnFyn) as a negative control, in which K299 was replaced with M. This point mutation was previously reported to be responsible for a lack of Fyn TK activity [38]. The WB analysis revealed that Fyn TK activity was inhibited because Tyr420 was less phosphorylated in dnFyn neurons (Figure 2C), thus resulting in a reduction of pAPP-Tyr levels in APP+dnFyn neurons compared with APP+Fyn co-transfected neurons (Figure 2C).

APP has three putative Tyr phosphorylation sites at the C-terminal domain (Tyr653, Tyr682, and Tyr687) [39]. To exclude the possibility that Tyr residues other than Tyr682 are phosphorylated by Fyn TK, we included APP YG control DNA, in which Tyr682 is replaced by Gly. Notably, pAPP-Tyr levels significantly decreased in YG+Fyn neurons compared with APP+Fyn neurons (Figure 2A), demonstrating that Fyn (either alone or upstream of the TK cascade) ultimately and preferentially promotes APP phosphorylation at the Tyr682 residue.

Transfected neurons were stained with DAPI and counted and the corresponding media were processed by ELISA for the measurement of Aβ42 levels. Data reported in Appendix A, showed that neurons transfected with APP+Fyn secreted a significant amount of Aβ42 in media (Appendix A) and largely die (Appendix A) when compared with APP neurons. Conversely, when APP neurons were co-transfected with dnFyn as well as when APP YG neurons were co-transfected with Fyn, the extent of Aβ42 in the media and the number of death neurons were not significantly different than APP neurons indicating that, when overactivated, Fyn promotes APP Tyr682 phosphorylation and triggers amyloidogenic processes and neuronal death. Consistently, WB analysis performed on total lysate from APP, Fyn and APP+Fyn neurons analyzed with anti-Aβ antibody 4G8 shows a large amount of Aβ structures in APP+Fyn neurons, further indicating the increased amyloidogenic APP processing upon Fyn overactivation conditions (Appendix A).

### 3.3. Fe65 Promoted Fyn Mediated APP Tyr682 Phosphorylation in Human Neurons

Fe65 mediates APP phosphorylation at Tyr682 and modulates APP processing and Aβ peptide generation [40,41,42,43,44]. Fe65 is a multidomain protein that includes an N-terminal α-helical domain and three protein-protein interaction modules (i.e., one WW domain and two consecutive C-terminal phosphotyrosine-binding (PTB) domains) [45,46,47,48]. The WW domain appears to be crucial for actin dynamics and cell motility and thus regulates the functions of developing neurons [49]. The PTB1 domain appears to be involved in binding APP, particularly the GYENPTY motif [50,51]. Fe65 also appears to be responsible for APP amyloidogenic processing [52,53,54].

Most TKs belong to the Src family. Fyn has a complex molecular structure that consists of a unique N-terminal sequence and three protein modules: Src-homology-3 (SH3) module (an interaction domain that is specialized for the recognition of xPxxP sequence motifs), Src-homology-2 (SH2) module, which recognizes pTyr, and the catalytic domain, which binds and cleaves adenosine triphosphate and mediates Tyr phosphorylation in protein targets. Intramolecular interactions that involve the SH3 and SH2 domains contribute to the negative regulation of Fyn kinase activity [36,54,55,56,57,58,59].

Fe65 was previously reported to bind the SH3 domain of Abl TK through its WW domain [60] and trigger APP phosphorylation at Tyr682 in cell culture [14,48]. Therefore, we hypothesized that Fe65 plays a potential role in facilitating Fyn-mediated APP phosphorylation at Tyr682 in AD neurons (Figure 3). Neurons were transfected with Fe65, Fe65+Fyn, Fe65+ΔSH3, and Fe65+mSH2, and the efficiency of transfection was assessed by WB (Figure 3A,B). Transfected neurons were then processed for pTyr IP and analyzed with anti-APP (Figure 3C,D). Fe65 alone increased endogenous pAPP-Tyr levels which were further increased in Fe65+Fyn neurons. Fe65+ΔSH3 and Fe65+mSH2 reduced pAPP-Tyr levels when compared with Fe65+Fyn neurons, suggesting that both motifs were important for Fe65-mediated pAPP-Tyr682. Notably, pAPP-Tyr682 levels were still detectable in Fe65+ΔSH3 and Fe65+mSH2 neurons, although at significantly lower levels compared with Fe65+Fyn neurons. This result may be attributable to the effect of Fe65 on endogenous Fyn and APP levels, suggesting that Fe65, similar to previous reports for Abl TK [14], might control APP phosphorylation at Tyr682 by interacting with SH3 and SH2 motifs of Fyn protein when overexpressed in neurons.

To further characterize the role of Fe65 in Fyn-mediated pAPP-Tyr682, we assessed pAPP-Tyr levels in Fe65+dnFyn neurons compared with Fe65 and Fe65+Fyn ΔSH3 neurons (Figure 4C,D). Fe65 increased endogenous pAPP-Tyr levels in Fe65 neurons. This increase was less evident in Fyn ΔSH3 (Fe65+FynΔSH3) or dnFyn (Fe65+dnFyn) neurons.

### 3.4. APP Tyr Phosphorylation Was Higher in Neurons from AD Patients as Compared with Neurons from Healthy Volunteers

Previous findings from brain tissues showed greatly enhanced pAPP-Tyr682 levels in reactive astrocytes of AD patients [61]. Therefore, we questioned whether pAPP-Tyr682 levels were also increased in neurons from AD patients compared to healthy volunteers. Patients were classified by age of sampling and gender. ApoE polymorphism status and the presence of AD-related mutations were included when possible (Table 1). After 5 weeks in culture, neurons were processed for IP using anti-phosphotyrosine (pTyr) antibody and analyzed by WB using an anti-APP antibody, based on previously described procedures [20,33] and reported in the Methods. We found that the levels of pAPP-Tyr were significantly higher in nine of the 10 analyzed AD patients (Figure 5A,C) compared with healthy volunteers and with the FTD patient. pAPP-Tyr levels in each patient were normalized relative to basal APP levels (Figure 5B,D). Notably, basal APP levels did not differ between AD patients, the FTD patient, and healthy volunteers (Figure 5D).

### 3.5. Fyn TK Activity Was Increased in Neurons from AD Patients as Compared to Healthy Volunteers

Neurons from AD patients and healthy volunteers were also processed for the determination of Fyn expression levels and Fyn TK activity. As shown in Figure 6A,B Fyn expression levels were not different among AD patients, the FTD patient, and healthy controls. However, an increase in the levels of phosphorylated Fyn at Tyr420 (pFyn-Tyr420) was detected in eight of nine neurons of AD patients and in the FTD patient compared with healthy donors, thus reflecting an increase in Fyn TK activity (Figure 6A,B).

### 3.6. APP Tyr Phosphorylation Was Decreased in Neurons in Which Fyn Expression Levels Were Knocked Down

To evaluate whether Fyn TK is responsible for the increase in pAPP-Tyr in neurons from AD patients, we inhibited Fyn expression levels using Fyn siRNA (see Methods) in neurons from three AD patients (patient no. 31F, 38F, and 53M) in which we previously reported a progressive increase of Aβ42 levels in media and neuronal death after 4 weeks in culture [33]. Neurons from the AD patients were all provided by Axol Bioscience. The use of materials from this single source minimized potential technical variables. A 50% reduction of Fyn expression (Figure 7A,C) was sufficient to reverse the increase in pFyn-Tyr420, pAPP-Tyr compared with corresponding unsilenced controls (31F vs. siFyn #31F; 38F vs. siFyn 38F; 53M vs. siFyn #53M; Figure 7B,D) suggesting that Fyn is involved in the increase in pAPP-Tyr in these AD neurons.

### 3.7. TKI Exposure Inhibited Aβ42 Release in Neurons from AD Patients with High APP Tyr Phosphorylation Levels

In order to evaluate whether Fyn might be a valuable strategy to reduce pAPP-Tyr and prevent Aβ42 production in neurons, we selected neurons from five patients, four diagnosed with AD (patient no. 87F, 31F, 38F, and 53M) and one diagnosed with FTD (patient no. 64F) all provided by Axol Bioscience. Fyn TK was activated in all four AD neurons and in FTD neurons. Among these, the patients nos. 31F, 38F, and 53M had elevated pAPP-Tyr, whereas patient no. 87F (AD) and patient no. 64F (FTD) had a low level of pAPP-Tyr (Figure 8). ELISA revealed Aβ42 secretion in the media of all AD neurons at a level that was both detectable and quantifiable. In contrast, Aβ42 levels in the FTD patient sample were barely above the limit of detection.

According to the manufacturer’s specifications, in order to inhibit Fyn TK activity, we used 10 nM saracatinib (IC_50_ = 0.7 nM), 10 nM dasatinib (IC_50_ = 0.8 nM), 150 nM SU6656 (IC_50_ = 170 nM), 150 nM PP2 (IC_50_ = 150 nM), and 100 nM masitinib (Table 2). Of note, we previously reported that sunitinib and PP2 inhibit pFyn-Tyr and pAPP-Tyr as well as restore APP trafficking from neurons of patients #31F, #38F, and #53M [20]. In all of the experiments, neurons were exposed to the TKIs for 6 h, and the medium was collected after an additional 12 h according to a previously described procedure [62]. Among the TKIs, saracatinib and SU6656 effectively inhibited Aβ42 release in neurons from the three AD patients that had high pAPP-Tyr levels (Figure 5A). Dasatinib, PP2, and masitinib were less effective at the selected concentrations, although they significantly reduced Aβ42 release in the three patients who had high pAPP-Tyr levels when compared to control neurons incubated with the vehicle. Interestingly, the TKIs did not significantly reduced Aβ42 levels in the AD patient (patient no. 87F) or in the FTD patient (patient no. 64F) who did not have high pAPP-Tyr levels.

## 4. Discussion

Intensive research has revealed some of the key pathogenetic mechanisms of AD and identified some promising biomarkers for diagnosis and prognosis, but further advances are still necessary. Highly variable genetic and environmental factors and the fact that the etiology of AD remains elusive despite intensive research efforts have hampered the ability to provide an early diagnosis of AD and complicated the development of effective therapies. Here, we sought to improve our understanding of the function of APP and in particular of its Tyr682 residue in neurons and the ways in which high Tyr682 phosphorylation levels lead to Aβ42 production, neurotoxicity, and ultimately AD.

APP is constitutively phosphorylated in brain tissues and neuronal cells [6,18]. Eight possible phosphorylation sites in the APP C-terminus have been identified to date. Seven of these sites are reported phosphorylated in AD brains (Tyr653, Tyr682, Tyr687, Thr668, Thr686, Ser655, and Ser675) [63]. Among the APP Tyr residues, we and others previously reported a crucial role for the Tyr682 residue in APP endocytosis and the ways in which alterations of pAPP-Tyr682 levels are related to AD onset and progression [6,11,18,64].

In the present study, we found that APP phosphorylation at Tyr increased in neurons from nine of 10 AD patients but not in healthy volunteers or in one patient with FTD, further substantiating the pathophysiological significance of APP Tyr phosphorylation as an early sign of AD and pointing to the therapeutic targeting of APP Tyr phosphorylation as a potential novel strategy in AD treatment.

Tyr682 can be phosphorylated by several mechanisms by different TKs although the functional meaning of this phosphorylation remains unclear. Indeed, among the other TK, c-Abl, and several TK belonging to the Src family have been reported to phosphorylate APP at Tyr682 in different cell lines [14,39,65], suggesting the involvement of more than one TK in this process. In addition, we previously reported that the nerve growth factor receptor TrkA is essential for APP phosphorylation at Tyr682 in the mouse brain [10,66].

In the present study, we found that Fyn phosphorylation at Tyr420 residue is largely increased in AD neurons in 8 out of 10 patients and in neurons from the FTD patient. Among these eight patients, six also showed increased pAPP-Tyr levels. The silencing of Fyn expression or the exposure to Fyn TKIs inhibited Fyn TK activity in neurons from three of these patients with high pAPP-Tyr levels and prevented Aβ42 release. Differently, neurons with increased pFyn-Tyr420, but not pAPP-Tyr, exposed to Fyn TKI, keep releasing Aβ42 in media. Of note, Fyn overexpression largely and selectively triggered APP phosphorylation at Tyr682 in healthy neurons and promoted APP processing to generate intracellular AICD and extracellular Aβ42, ultimately leading to neuronal death. Altogether these results underline the role of Fyn TK in mediating APP Tyr682 phosphorylation. Although the lack of a procedure that allows the selective detection of pAPP-Tyr682 in neurons insinuates that Fyn might influence the phosphorylation also of other Tyr(s) on the APP sequence rather than Tyr682, these results clearly emphasize the upstream role of APP Tyr682 phosphorylation in the APP amyloidogenic processing and the possibility to targeting Fyn for the development of new therapeutic strategies in AD.

The inhibition of TK as a potential strategy in the treatment of AD as well as of other neurodegenerative diseases has been previously suggested. In animal studies, TKI-induced inhibition of the TK Abl increases Aβ clearance, and attenuates behavioral deficits [67]. Imatinib, a small-molecule TKI, ameliorates neurodegenerative changes in animal models of AD through the transcriptional induction of genes that are involved in the clearance of Aβ fibrils, in the regulation of apoptosis, and in the decrease of pathological tau protein hyperphosphorylation [65,67,68]. Other TKIs, such as nilotinib and bosutinib, can efficiently cross the blood-brain barrier and have been proposed to be potential treatments for AD and Parkinson’s disease [69,70]. Preclinical treatment studies reported that saracatinib blocks Fyn expression in brain slices and rescues synaptic depletion and spatial memory deficits in APP/PSEN1 mice [35,36,71,72]. Saracatinib and masitinib are currently being tested in phase II and phase III clinical trials for mild-to-moderate AD (NCT01864655, NCT02167256, NCT00976118, and NCT01872598). In the NCT00976118 trial, oral masitinib was administered over a period of 24 weeks concomitantly with an acetylcholinesterase inhibitor (donepezil, rivastigmine, or galantamine) or memantine [69]. However, despite a large number of trials in progress, the use of TKI as a treatment for AD patients is still questionable.

Among the limited number of patients who were included in this study, we found significant improvements in terms of neuronal survival and Aβ42 release in AD neurons exposed to Fyn TKIs. However, such effects were only detectable in patients with high levels of APP Tyr phosphorylation, suggesting that the treatment with Fyn TKIs may yield better results if limited to those patients in which the levels of APP Tyr682 phosphorylation are elevated, thus reducing off-target side effects and improving the clinical outcome. In this context, the level of pAPP-Tyr682 might help identify individuals who will experience the most drug benefits and fewest side effects moving towards the use of specific Fyn TKI therapies targeted to each patient. Indeed, our results also support the importance of developing diagnostic procedures to make pAPP-Tyr682 levels detectable in peripheral tissues (e.g., blood) as an AD biomarker.

Fyn-mediated APP phosphorylation at Tyr682 requires other players. Tyr682 phosphorylation controls the binding of APP to specific adaptor proteins, including Fe65, Numb, Numb-like, Jip1, Shc, and X11α. These adaptor proteins specifically recognize and fold the AICD [18] and indirectly modulate Aβ production. These APP-interacting proteins are characterized by having at least one (two in the case of Fe65) PTB domain. This module directly binds to the _682_YENPTY_687_ motif and other proteins (e.g., TK) and controls APP phosphorylation at Tyr682. In vitro studies have implicated more than 20 adaptor proteins that can interact with the AICD [51], but remaining unknown is whether these interactions occur in vivo and which of them are relevant to APP functions and AD. Among them, Fe65 is one of the most studied adaptor proteins. Because of the presence of two PTB domains, the Fe65-AICD interaction spans almost the entire AICD-C31 fragment [73] and thus shapes the APP structure and induces the APP phosphorylation at Tyr682 and AICD production.

In the present study, we found that APP phosphorylation at Tyr682 increased in neurons that overexpressed Fe65, thus indicating that Fe65 allows TK to interact with APP. Interestingly, such phosphorylation was almost completely abolished in dnFyn neurons, demonstrating that Fyn is involved in the Fe65 mediated mechanism of APP Tyr682 phosphorylation. Similarly, Fe65 capability to phosphorylate APP at Tyr682 is largely reduced in neurons that lacked the Fyn docking site for binding to proteins that contain the xPxxP motif (Fyn ΔSH3), such as Fe65, as well as in neurons carrying a mutation of the Fyn docking site to Fe65 (Fyn mSH2), likely indicating that Fe65 mediated APP Tyr682 phosphorylation requires the interaction with either the Fyn SH3 or Fyn SH2 motif. Interestingly, the fact that the ability to phosphorylate APP at Tyr682 is largely reduced, but not totally lost, in Fe65 overexpressing neurons in presence of dnFyn, Fyn+ΔSH3, and Fyn+mSH2, outlooks the possibility that Fyn is part of a more complex scenario that might involve other TKs. Indeed, despite all these aspects deserving further investigation, our findings suggest a crucial upstream role for Fyn and Fe65 and their clinical relevance as a possible pharmacological target in controlling APP Tyr682 phosphorylation and thus preventing Aβ42 increase in AD neurons.

## 5. Conclusions

In conclusion, these findings point on APP Tyr682 phosphorylation as a valuable strategy for AD treatment and that the development of selective Fyn TKIs might be a promising option in the management of AD.

Studies are in progress in our team to develop a routine diagnostic procedure allowing the determination of the pAPP-Tyr682 phosphorylation status in peripheral tissues that might provide a new diagnostic strategy that will allow early detection of the pathology before Aβ42 accumulates and initiates neurodegenerative processes.

## Figures and Tables

**Figure 1 cells-09-01807-f001:**
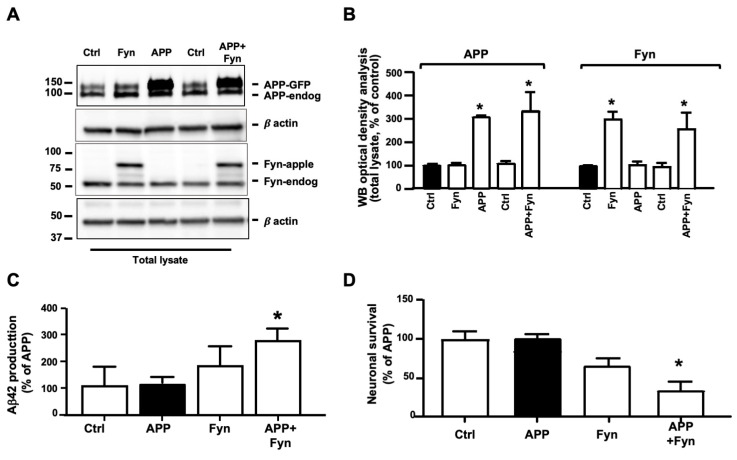
Amyloidogenic amyloid precursor protein (APP) processing increased in APP-GFP- and Fyn-apple-tagged neurons. (**A**) APP, Fyn, and β-actin Western blot (WB) analysis of neural stem cells after 48 h of transfection with APP-GFP (500 ng, APP) and Fyn-apple (500 ng, Fyn). A GFP empty vector was used as an internal loading control (Ctrl) or to balance the total amount of DNA for each experimental point (1 µg DNA/well). Total lysate (20 µg) from each sample was loaded on 4–20% Tris-Gly. The densitometric analysis of APP-GFP and Fyn-apple is reported in (**B**). Optical density values were normalized to β-actin and were expressed as a percentage of Ctrl (black bar). *n* = 3. * *p* < 0.05, vs. control using one-way ANOVA followed by Dunnett’s post hoc test. (**C**) ELISA analysis of Aβ42 from media of Ctrl, APP-GFP, Fyn-apple, and APP-GFP+Fyn-apple transfected neurons. Aβ42 levels were normalized to the number of alive cells (DAPI stained nuclei) that were present on each slide after 48 h of transfection and were expressed as ng of Aβ42 that were released from each cell in 0.5 mL of media. Data are reported in (**C**) as % of APP transfected neurons. Each experiment was performed three times in triplicate (*n* = 3). * *p* ≤ 0.05, vs. APP (black bar). Statistically significant differences were calculated using one-way ANOVA followed by Dunnett’s post hoc test. (**D**) Transfected neurons and GFP transfected controls were stained with DAPI, and the number of alive neurons was counted under an immunofluorescence microscope. Each experiment was performed three times in triplicate (*n* = 3). The data are expressed as a percentage of APP. * *p* < 0.05, vs. APP (black bar). Statistically significant differences were calculated using one-way ANOVA followed by Dunnett’s post hoc test.

**Figure 2 cells-09-01807-f002:**
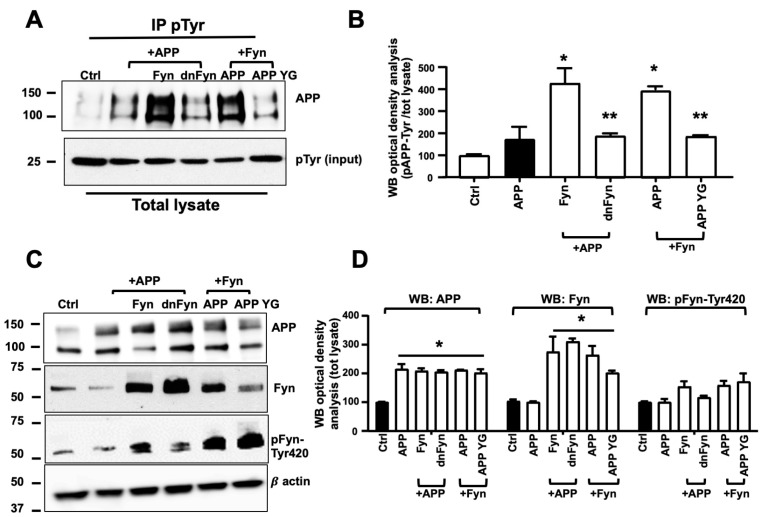
Fyn triggered APP phosphorylation at Tyr682 in human neurons. (**A**) Untransfected control neurons (Ctrl) and APP, APP+Fyn, APP+dnFyn, and APP YG+Fyn were immunoprecipitated with mouse anti-pTyr magnetic beads and analyzed by WB with rabbit anti-APP antibody. WB densitometric analysis is reported in (**B**). Values were calculated by dividing pAPP-Tyr by the corresponding APP optical density values (pAPP-Tyr/APP). Statistically significant differences were calculated using one-way ANOVA followed by Dunnett’s post hoc test. *n* = 3. * *p* ≤ 0.05, vs. APP; ** *p* < 0.05, vs. Fyn. (**C**) WB analysis of APP, Fyn, pFyn-Tyr420, and β-actin from total lysates of untransfected control neurons (Ctrl) and APP, APP+Fyn, APP+dnFyn, and APP YG+Fyn. pFyn-Tyr420 levels were calculated as a ratio of pFyn-Tyr420 relative to the corresponding Fyn optical density values (pFyn-Tyr420/Fyn). The densitometric analysis is reported in (**D**). The data are expressed as a percentage of untransfected control (empty vector). Statistically significant differences were calculated using one-way ANOVA followed by Dunnett’s post hoc test. *n* = 3. * *p* < 0.05, vs. control (black bar).

**Figure 3 cells-09-01807-f003:**
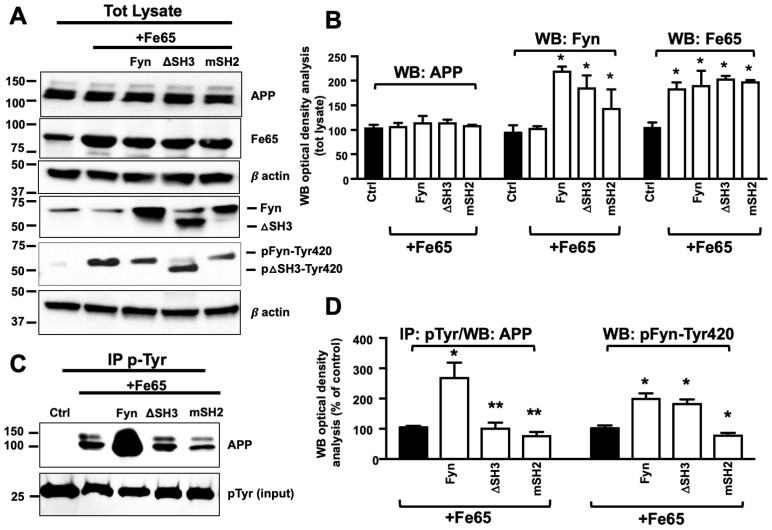
SH2 and SH3 motifs were crucial for Fyn-mediated APP phosphorylation at Tyr in Fe65-transfected neurons. (**A**) WB analysis of APP, Fyn, pFyn-Tyr420, and Fe65 proteins in total lysates from neural stem cells that were transfected with Fe65 with and without Fyn ΔSH3, and Fyn mSH2. The densitometric analysis of APP, Fyn, and Fe65 expression before and after transfection is reported in (**B**). Densitometric analysis of pFyn-Tyr420 is reported in (**D**) instead. The data were normalized to β-actin values and are expressed as a percentage of control (Ctrl, empty vector). *n* = 3. * *p* < 0.05, vs. Ctrl. (**C**) Samples were immunoprecipitated with mouse anti-pTyr magnetic beads and analyzed by WB using the rabbit anti-APP antibody. The optical density analysis is reported in (**D**). Values were calculated by dividing pAPP-Tyr or pFyn-Tyr optical density values by the corresponding APP or Fyn values in total lysates, respectively. * *p* ≤ 0.05, vs. Fe65; ** *p* ≤ 0.05, vs. Fe65+Fyn. *n* = 3. Statistically significant differences were calculated using one-way ANOVA followed by Dunnett’s post hoc test.

**Figure 4 cells-09-01807-f004:**
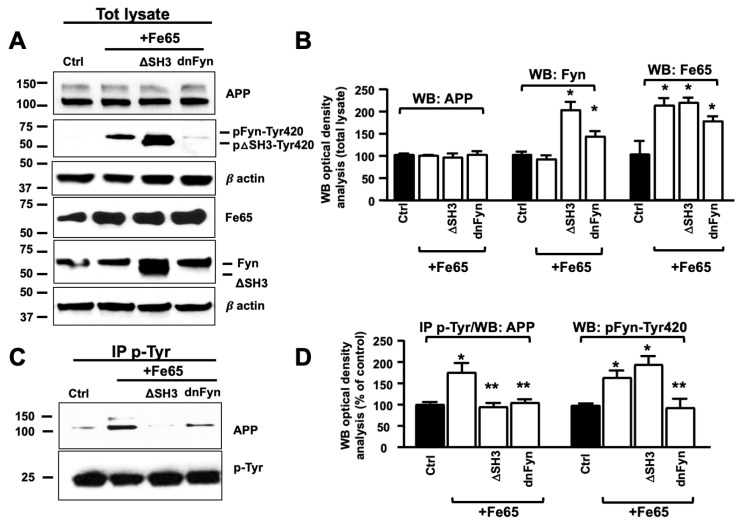
Fe65 controlled Fyn-mediated APP phosphorylation at Tyr. (**A**) WB analysis of APP, Fyn, pFyn-Tyr420, and Fe65 proteins in total lysates from Fe65, Fe65+Fyn, Fe65+ΔSH3, and Fe65+dnFyn neurons. The densitometric analysis is reported in (**B**). Densitometric analysis of pFyn-Tyr420 is reported in (**D**) instead. The data were normalized to β-actin values and were expressed as a percentage of Ctrl (empty vector). *n* = 3. * *p* < 0.05, vs. control. (**C**) Samples were immunoprecipitated with mouse anti-pTyr magnetic beads and analyzed by WB using rabbit anti-APP or rabbit anti-Fyn antibody. pAPP-Tyr optical density analysis is reported in (**D**). Values were calculated by dividing pAPP-Tyr or pFyn-Tyr420 optical density values by the corresponding APP or Fyn optical density values in total lysates, respectively. Statistically significant differences were calculated using one-way ANOVA followed by Dunnett’s post hoc test. *n* = 3. * *p* ≤ 0.05, vs. control; ** *p* ≤ 0.05, vs. Fe65+Fyn.

**Figure 5 cells-09-01807-f005:**
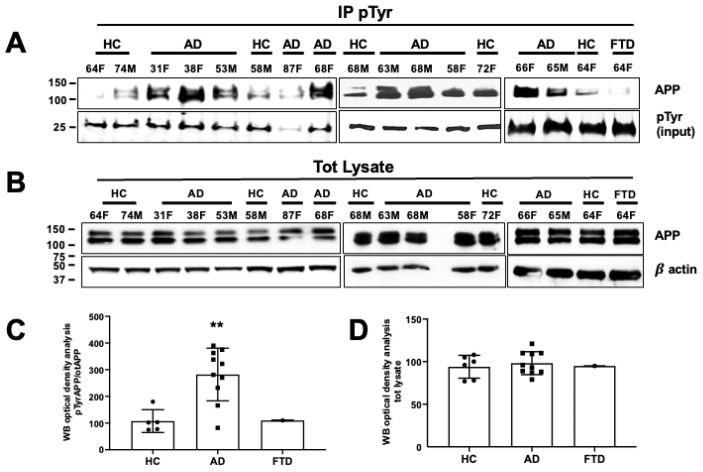
The amyloid precursor protein was phosphorylated at the Tyr residue in neurons from AD patients. (**A**) After 5 weeks in differentiating media, neurons from healthy controls (HCs) and AD patients and one FTD patient were processed for pTyr IP using mouse anti-pTyr magnetic beads- conjugated antibody (4G10) and analyzed by WB using rabbit anti-APP antibody (Y188). Membranes were blotted with anti-mouse IgG, and the pTyr band (input) migrating at 25 kDa was used as a loading control. (**B**) WB analysis of basal APP levels using the rabbit anti-APP antibody (Y188). β-actin was used as a loading control. (**C**,**D**) Densitometric analysis of pAPP-Tyr levels in AD patients vs. HCs (**C**) and total APP levels expressed as a percentage of HCs (**D**). Mean optical density values were calculated as the ratio of pAPP-Tyr levels relative to basal APP levels (after normalization to β-actin) from each sample (experiments from each sample were repeated three times, *n* = 3). Statistically significant differences were calculated using Student’s *t*-test. ** *p* < 0.01 vs. HCs.

**Figure 6 cells-09-01807-f006:**
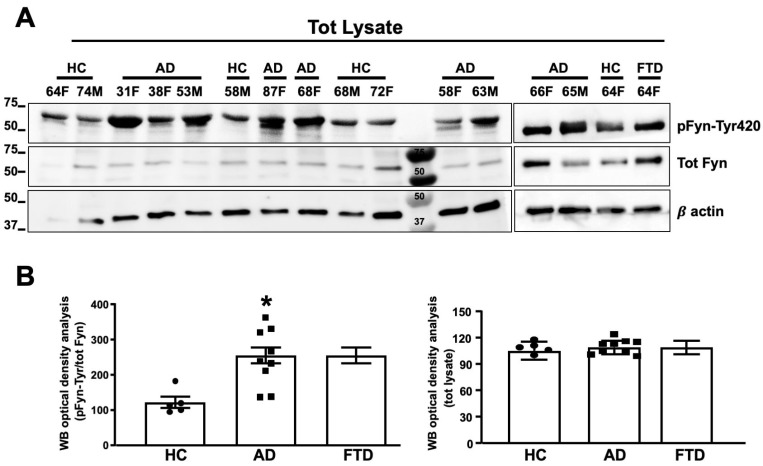
Fyn was activated in neurons from AD patients. (**A**) Fyn TK activity, expressed as an increase in pFyn-Tyr420 relative to total Fyn, was assessed by WB using the rabbit anti-pFyn-Tyr420 antibody and rabbit anti-Fyn antibody. The increase in Fyn TK activity was calculated as a ratio of pFyn-Tyr420 levels relative to basal Fyn levels and expressed as % of healthy control (HC). The optical density analysis of pFyn-Tyr420/Fyn is shown in (**B**). Basal Fyn levels were normalized to β-actin and are expressed as a percentage of HC. Experiments from each sample were repeated three times, *n* = 3). * *p* < 0.05 vs. HC. Statistically significant differences were calculated using Student’s t-test. Note that the experiments reported in Figure 5 and Figure 6 were performed blind and HC 64F has been loaded two times (right and left panels).

**Figure 7 cells-09-01807-f007:**
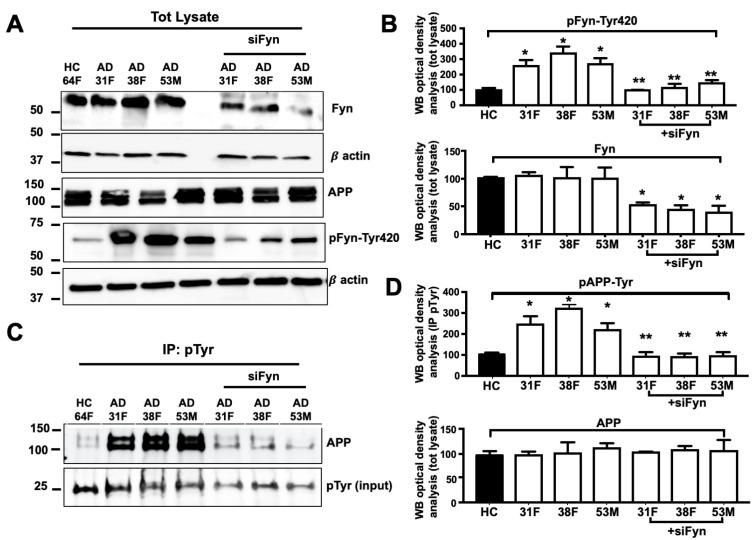
pAPP-Tyr levels decreased in Fyn knockout neurons from AD patients. (**A**) WB analysis of neural stem cells from one healthy control (HC, patient no. 64F) and three AD patients (the patient no. 31F, 38F, and 53M) before and after Fyn RNA silencing (siFyn). Forty-eight hours after Fyn siRNA transfection, neurons were collected and processed for WB using rabbit anti-Fyn, rabbit anti-Fyn pTyr420 (Src-pTyr416), rabbit anti-APP, and β-actin antibodies. Densitometric WB analysis is reported in (**B**). *n* = 3. * *p* < 0.05, vs. HC (one-way ANOVA followed by Dunnett’s post hoc test). (**C**) Total lysates were immunoprecipitated with mouse anti-pTyr magnetic beads (IP pTyr) and analyzed by WB using the rabbit anti-APP antibody. The optical density analysis is reported in (**D**). Values were calculated as a ratio of pAPP-Tyr relative to APP. pTyr levels were used as loading controls. *n* = 3. * *p* ≤ 0.05, vs. HC; ** *p* < 0.05, vs. the corresponding unsilenced neurons from AD patients (one-way ANOVA followed by Dunnett’s post hoc test).

**Figure 8 cells-09-01807-f008:**
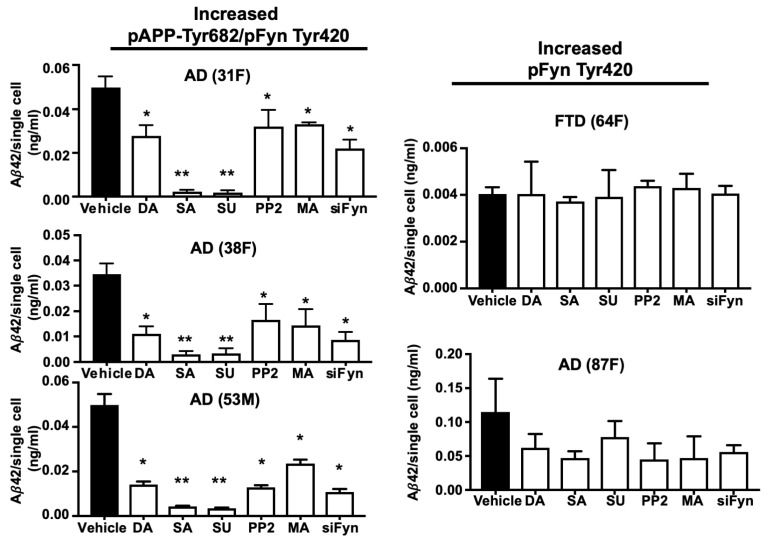
Fyn TKIs reduced Aβ42 levels in neurons from AD patients who had high pAPP-Tyr682 levels but not in those in which pAPP-Tyr682 levels were not altered. After 5 weeks in culture, neurons were exposed to Fyn TKIs for 6 h and then left for an additional 12 h in fresh media without inhibitors, based on a previously described procedure [12]. The Fyn TKIs used in the experiments are reported in Table 2. All the inhibitors were resuspended in the same vehicle (DMSO). The following concentrations of each inhibitor were used: 10 nM saracatinib (SA), 10 nM dasatinib (DA), 150 nM SU6656 (SU), 150 nM PP2, and 100 nM masitinib (MA). Aβ 42 levels from neurons transfected with siFyn are also reported. Aβ42 was assessed in media by ELISA. Neurons were stained with DAPI and counted. No significant alterations in neuronal survival were assessed in the presence, or not, of inhibitors (data not shown). Aβ42 values (ng/mL) were calculated as a ratio of total Aβ42 levels relative to the total number of alive neurons and are expressed as Aβ42 levels that were released from a single cell in 0.5 mL of media. *n* = 3 * *p* < 0.05, vs. vehicle; ** *p* < 0.05, vs. DA. Statistical significance was calculated using one-way ANOVA followed by Dunnett’s post hoc test.

**Table 1 cells-09-01807-t001:** Features of neural stem cells and induced pluripotent stem cells (IPSC) from Axol Bioscience and Coriell Institute, respectively. Age and gender are shown for each patient. ApoE genotype and the presence of AD-related mutations (presenilin 1, PS1 or microtubule associated protein tau, MAPT) in donors were included when available. Aβ levels in cerebrospinal fluid from AD patients from Coriell Institute were reported according to information that is available on the website.

**Neural Stem Cells**					
	**Sampling** **Age**	**Gender**	**ApoE** **Genotype**	**AD-Related** **Mutation**	**Source** **(#)**
**Healthy** **Controls** **(HC)**	64	F	------		Ax0019
	74	M	E2/E2		Ax0018
	Cord blood	-----	------		Ax0015
**AD** **Patients** **(AD)**	38	F	E3/E3	PS1 L286V	Ax112
	53	M	------	PS1 M146L	Ax113
	31	F	E3/E4	PS1 A246E	Ax114
	87	F	E4/E4	------	Ax111
**Frontotemporal Dementia (FTD)**	64	F	------	MAPT P301L	Ax324
**IPSC**					
	**Sampling** **Age**	**Gender**	**ApoE4** **carrier**	**CSF Aβ Presence**	**Source (#)**
**Healthy** **Controls** **(HC)**	58	M			CW70073
	72 (diabetic)	F			CW50068
	68	M			CW50028
**AD** **Patients** **(AD)**	68	F	Yes	Yes	CW50025
	68	M	Yes	Yes	CW50082
	58	F	Yes	Yes	CW50018
	63	M	No	Yes	CW50024
	66	F	----	Yes	CW50069
	65	M	----	Yes	CW50126

**Table 2 cells-09-01807-t002:** Tyrosine kinase inhibitors. Commercial name of TKIs and their abbreviations are reported on the left. Fyn TK IC50 is listed on the right.

IC50 of Tyrosine Kinase Inhibitors Active on Fyn	
Dasatinib (DA)	0.8 nM
Saracatinib (SA)	2.7 nM
SU6656 (SU)	130 nM
PP2	4 nM / 5 nM
Masitinib (MA)	10 nM

Source Selleckchem https://www.selleckchem.com/.

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
