# Peer review of "Fyn Tyrosine Kinase Elicits Amyloid Precursor Protein Tyr682 Phosphorylation in Neurons from Alzheimer’s Disease Patients"

_cells, 2020, doi:10.3390/cells9081807_

Round 1

Reviewer 1 Report

The determination of the pAPP-Tyr682 phosphorylation status in peripheral tissues for an early detection and/or pharmacological targeting of the AD pathology is definitely a subject of great interest for those involved in basic and clinical research on neurodegenerative diseases.

Authors should explain and justify the use of Human induced pluripotent stem cells (iPSCs) and human Neural stem cells (hNSCs).

Ethical statement and considerations should be made regarding the use of those particular cells according to the rules and legislation of Denmark, Italy and Germany

Authors should explain the following facts:

  • Figure 5B. APP bands from the eight samples on the left are saturated and protein levels of APP cannot be quantified by densitometric analysis
  • Figure 6A. There is no band of β-actin from HC 64F

Author Response

We thank Reviewer 1 for his/her comments.

Figure 5B has been replaced in this revised version

In the regard to the ß-actin of “HC 64F”, the same sample is loaded together with FTD sample on the right panel of Fig6A. A note to clarify this aspect has been added in the legend of Fig6 in this revised version of the manuscript.

Either IPSC or NSC are commercially available and we bought them from Coriell Institute or Axol Bioscience, respectively.

IPSC cells were induced to NSC and differentiated for 5-7weeks in order to obtain mature neurons. Only iPSC showing a mature neuronal phenotype were included in the study. Unfortunately, Axol Bioscience only commercializes 5 NSC lines and we needed a larger number of samples to give consistency to our hypothesis and to support our evidence. This is the reason because we included both NSC and IPSC lines (differentiated in neurons) in our study.

Reviewer 2 Report

Mounting evidence has shown that Tyr682 phosphorylation is an important signal leading to the generation of Abeta42. Therefore, identifying a specific inhibitor of T668 phosphorylation might be the target of AD therapy (Chang et al 2006, Mol Cell Biol).

Fyn tyrosine kinase (TK) interaction with pAPP-Tyr682 has make it an attractive therapeutic target. Here, the authors showed that inhibiting Fyn TK activity abolished pAPP-Tyr882 phosphorylation and reduced Abeta42 secretion in cellular models. The team has also included evidence showing Fe65 is a mediator for pAPP-Tyr682 phosphorylation. Taken together, targeting the Fe65-APP-Fyn pathway could regulate the progression of AD pathology.

While this is not a new proposal (van der Kant and Goldstein, 2015 Dev Cell), the results in this study further consolidate the notion that targeting this signaling pathway is an innovative strategy to arrest AD progression.

I have the following queries;

(1) AD and FTD are pathologically distinct and hence good to separate the two diseases in figure 5C and 5D, and 6B and 6C. Since there is only 1 FTD case, this could be presented as a stand-alone bar graph.

(2) There are at least ten phosphorylated sites on APP that have been identified. They include two sites in the ectodomain (Ser198 and Ser206) and eight sites in the cytoplasmic domain (Tyr653, Tyr682, Tyr687, Ser655, Ser675, Thr654, Thr668 and Thr686 (Wang et al, 2017 Front Mol Neurosci). Has Fyn TK inhibition been reported not to affect the phosphorylation of other APP sites?

Author Response

We thank to Reviewer 2 for pointing on these relevant aspects 

In this revised version, we included data from FTD sample in 4 independent panels in 5C and 5D, as well as in 6B and 6C.

We agree with the reviewer about this aspect, in fact in the previous version of the manuscript we included a sentence in the results (lines 226-231) and shortly mentioned this aspect in the discussion (Lines 469-473). Indeed, Fyn might trigger the phosphorylation also of other APP Tyr residues, however the fact that APP YG mutation prevents Fyn mediated APP Tyr phosphorylation strongly suggests that this event is upstream and precedes any other phosphorylation of APP Tyr residues.

Reviewer 3 Report

The overall hypothesis of the study seems impressive but there are some issues with data representation as well as the coherence in the manuscript. Some suggestions and queries given below might help in improving the quality of the manuscript.

  1. All the reagents used can be mentioned under a separate heading in material and method section to maintain overall cohesiveness in the actual experimental methods used.
  2. In figure 2C it’s difficult to understand the western blots for non-phosphorylated form of Fyn and pFyn-Tyr420. There doesn’t seem to be a correlation between the two. Additionally the bands for pFyn-Tyr420 look somewhat inconsistent in the sense that somewhere you see double bands and somewhere it’s a very thick single band. Can the authors check if there’s some discrepancy in representing the data or if possible provide the original blots for the same?
  3. For Figure 3A can the authors provide another representative image for β-actin from the triplicate set of blots?
  4. In figures 5(A-B) and 6A the samples have been loaded rather randomly. It would have been better to load all healthy controls together followed by patient samples since it’s making the results very confusing to look at and understand.
  5. The APP bands for figure 5B are fused in some and separated in others. Is that due to differences in membrane processing and visualization?
  6. The manuscript needs a thorough grammatical proofread.

Author Response

We thank Reviewer 3 for highlighting these relevant aspects

  1. A separate heading called Materials has been included in this new revised version.
  2. We agree with the referee and we believe that this apparent inconsistency depends on the fact that pFyn antibody also recognizes other proteins of the Src family which Fyn belongs to. This aspect has been mentioned previously in Methods and is now in Materials (line 160). During the submission process I believe we uploaded all the original figures.
  3. Figure 3A has been replaced
  4. Experiments have been performed blind; this is the reason because samples appear to be randomly loaded.
  5. Samples have been collected in different times depending on their availability. They grow differently and display differences in the total proteins amount that in turn affects the way in which they migrate on the gel, this might probably answer your concerns. However, in this new version, we replaced 5A with one in which bands appear to be better separated.
  6. We apologize for some inaccuracies in the English that have been edited in this revised version

Round 2

Reviewer 3 Report

The reviewer thanks the authors for incorporating necessary changes in the manuscript based on the suggestions given earlier.